# The efficiency of rotavirus A spread to extraintestinal tissues is not determined by the levels of its replication in the gut

Sergei A. Raev, Maryssa K. Kick, Maria Chellis, Linda J. Saif, Talita P. Resende, Anastasia N. Vlasova ⓘ *

Center for Food Animal Health, Department of Animal Sciences, College of Food, Agricultural and Environmental Sciences, The Ohio State University, Wooster, Ohio, United States of America,

\* vlasova.1@osu.edu

## Abstract

Rotavirus A (RVA) spreads to multiple extraintestinal organs; however, it is not well understood what viral or host characteristics regulate the efficiency of this spread. We conducted this study to determine whether more efficient intestinal RVA replication leads to a higher rate of its extraintestinal spread. We also examined the distribution of known RVA glycan receptors in different tissues to better understand their potential role in facilitating viral dissemination to extraintestinal sites. We inoculated germ-free pigs with porcine OSU G5P[7] characterized by remarkably robust *in vitro/in vivo* replication, and G9P[13] which replicates to low-to-moderate titers and several other strains. Significantly higher RVA titers were observed in intestinal tissue/contents/feces of pigs infected with G5P[7], whereas G9P[13] was associated with a relatively modest intestinal replication but the most efficient extraintestinal spread. As expected, and coinciding with the increased sialic acid/glycan abundance and diversity, all RVA strains replicated to highest titers in the gut. Further, among the examined extraintestinal tissues, the lungs: a) had the highest frequency of RVA RNA detection; b) exhibited the highest host glycan diversity/abundance; and c) represented the only extraintestinal tissue in which both gross and microscopic lesions were observed. This further underscores the association between the RVA receptor diversity and respiratory lesions. This is the first experimental evidence that RVA extraintestinal spread does not depend on its replication efficiency in the gut. Additionally, these findings may provide an explanation for the current global dominance of G9P[13] and related RVA strains, which could be capable of airborne spread.

## Author summary

Rotavirus A (RVA) remains a major cause of severe gastroenteritis in animals, including pigs, especially when co-infections with enteric bacteria occur, leading

**Data availability statement:** All relevant data are within the manuscript and its supporting information files.

**Funding:** This work was supported by the Swine Health Information Center (24-082 SHIC to A.V.). The funders had no role in study design, data collection and analysis, decision to publish, or preparation of the manuscript.

**Competing interests:** The authors have declared that no competing interests exist.

to diarrhea, dehydration, and even death. While traditionally considered to be an enteric pathogen, RVA has been detected in extraintestinal tissues, and some studies suggest it may contribute to pathological changes beyond the gut. Given the strain-specific differences in RVA replication and receptor recognition, there is a critical need to investigate the mechanisms driving systemic dissemination. Our study addresses this gap by identifying potential factors involved in RVA escape from the intestine, thereby advancing our understanding of rotavirus pathogenesis in extraintestinal organs.

## Introduction

Replication of rotavirus A (RVA) in mature, terminally differentiated intestinal epithelial cells (IECs), primarily of the ileum and jejunum, leads to severe diarrhea, dehydration, and even death [1–3]. In addition, RVA is known to spread to some extraintestinal tissues such as lungs, salivary glands, pancreas, spleen, etc. More specifically, studies have shown the detection of RVA RNA in nasal swabs from pigs infected with human RVA [4,5], viral replication in salivary glands from mice and pigs [5,6], liver and lungs from pigs [7,8], and pancreas, heart and other rat tissues [9]. Several studies have demonstrated the strain-specific effects of RVA extraintestinal spread [9–11]. Considering that RVA strains replicate to variable levels in the gut [12,13], we hypothesized that strains with higher levels of intestinal replication would possess an increased ability to disseminate to extraintestinal tissues compared to those with lower replication levels.

While systemic dissemination of RVA may be attributed to the blood or lymphatic system [4,14], its replication in tissues outside the intestine likely depends on the presence of RVA receptors. The ability of RVA to interact with IECs is associated with the presence of a diverse group of RVA receptors, including cellular sialylated and non-sialylated glycans, integrins and heat shock cognate protein 70 [3]. However, only limited data are available regarding the presence and distribution of RVA receptors in porcine extraintestinal tissues except for sialic acids (SAs), whose region-specific distribution in lungs has been shown to contribute to replication of influenza A viruses [15,16]. In this study, we evaluated the expression profiles of several SAs, histo-blood group antigens (HBGAs), and other non-sialylated glycans known to serve as RV receptors [17]. The interaction of RVA with sialic acids (SAs) is a key determinant of strain-specific replication [18–20]. Our previous studies demonstrated that neuraminidase treatment of porcine ileal enteroids reduced replication of the G5P[7] strain, whereas the same treatment prior to infection with G9P[13] led to enhanced replication [13]. Differences in binding affinity for sialylated and non-sialylated glycans have also been observed in other RVA and RVC strains [18,19,21]. Thus, our secondary hypothesis was that RVA replication in extraintestinal tissues may be associated with the diversity of sialylated and non-sialylated glycans.

While some attempts have been made to link RVA replication outside the intestine to histological lesions in pigs, mice, and rats, definitive evidence directly implicating

RVA as the cause of extraintestinal tissue damage remains limited [7,9,22,23]. To eliminate potential confounding effects from other bacterial or viral infections and to study the association between RVA and pathology in extraintestinal tissues, we used a germ-free (GF) pig model for our experiments.

## Results

### Rotavirus strains exhibit varying replication levels in the small intestine

Consistent with the past observations, significantly higher titers of OSU G5P[7] compared to all other RVA strains were observed in intestinal contents (Figs 1A, 1B, S1A and S1B) and rectal swabs (Figs 1C and S1C). Replication levels of the other strains did not differ significantly between one another.

### Extraintestinal replication of rotavirus: Tissue-specific and strain-specific analysis

To analyze the permissiveness of different tissues for RVA replication, we performed a tissue-specific analysis (Fig 2 and S1 Data). RT-qPCR demonstrated that the RVA strains were present in various extraintestinal tissues (Figs 2A and S2A). RVA RNA was detectable in a variety of extraintestinal tissues as early as dpi 1 (Fig 2A). RVA RNA levels varied significantly, reaching up to $1 \times 10^5$ genome equivalents (GE)/ml in the liver at 4 days post-infection (dpi) following G9P[13] inoculation. When combining data from all four RVA strains used in this study, the lungs, blood, and liver exhibited the highest frequencies of RVA RNA detection, with 75.0% (39/52), 71.1% (37/52), and 63.5% (33/52) of samples testing positive, respectively. This was followed by the spleen (55.8%, 29/52) and the salivary glands (42.3%, 22/52) for RVA RNA detection by RT-qPCR.

Significantly higher RVA RNA levels of G9P[13] were observed in liver at dpi 3 compared to salivary glands ($p < 0.05$) (Fig 2A and 2B). This trend persisted until dpi 4, when a significantly higher level of G9P[13] was detected in the liver compared to the lungs, salivary glands, spleen, and blood ($p < 0.001$). For G5P[7] a significantly higher RVA RNA level was observed in lungs compared to all other tissues ($p < 0.001$) at dpi 4 (Fig 2B).

The analysis of RVA RNA detection frequency (Fig 2C) showed that similar to qRT-PCR data, the most prominent differences between tissues were observed at dpi 3–4 (Fig 2C and 2D): at dpi 3 liver showed a significantly higher frequency of G5P[7] RNA detection compared to the salivary glands and blood ($p < 0.05$); at 4 dpi, the salivary glands exhibited a significantly lower frequency of RVA RNA detection compared to the lungs and blood ($p < 0.05$); at 4 dpi the spleen

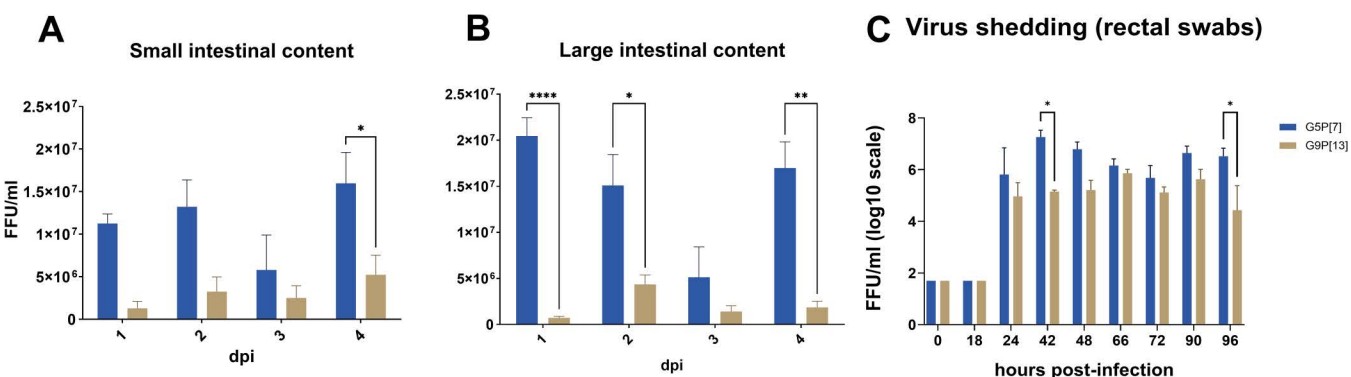

**Fig 1. Porcine RVA (G5P[7], G9P[13]) levels in the small intestine (A) and large intestine (B); RVA shedding (rectal swabs) (C).** Six-day-old germ-free pigs were orally inoculated with $1 \times 10^6$ FFU of each RVA. Swabs were collected at designated time points (day post-infection, dpi, 1-4). For the small and large intestine contents pigs were euthanized at the post-inoculation times indicated and contents were collected. RVA quantification was performed using cell culture immunofluorescence (CCIF). Significant differences (*$p < 0.05$, ** $p < 0.01$, *** $p < 0.001$) are indicated as calculated by using two-way ANOVA with repeated measures and the Tukey-Kramer test for multiple comparisons.

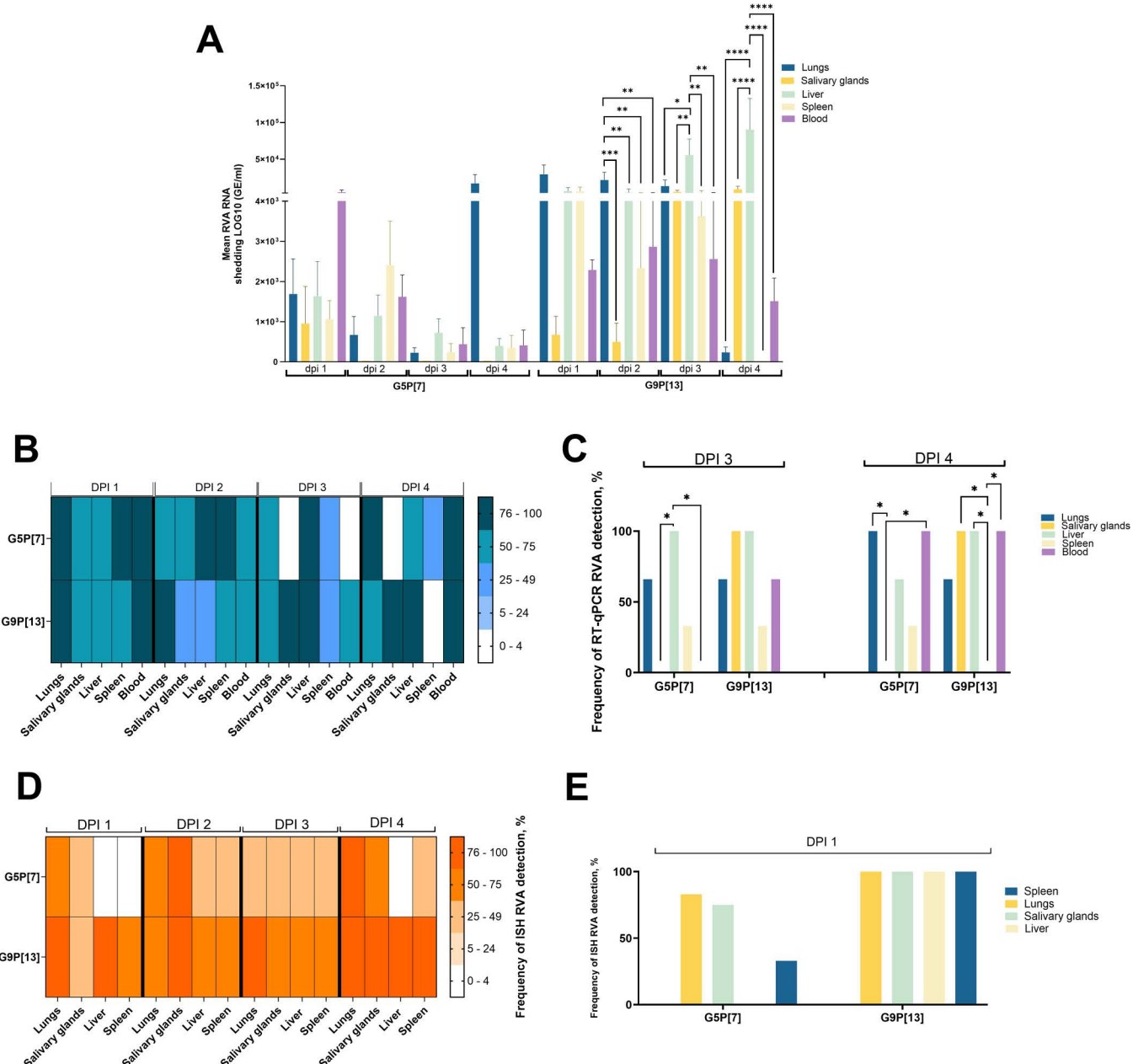

**Fig 2. Tissue-specific analysis of RVA RNA levels (G5P[7], G9P[13]) in extraintestinal tissues.** Six-day-old germ-free pigs were orally inoculated with $1 \times 10^6$ FFU of RVA. At the indicated post-inoculation time points, animals were euthanized, and tissues (lungs, salivary glands, liver, blood, and spleen) were collected. **(A)** RVA RNA levels in extraintestinal tissues across dpi 1–4. **(B)** RVA RNA levels in extraintestinal tissues at time points when significant differences were observed. Significant differences (*$p < 0.05$, ***$p < 0.001$, ****$p < 0.0001$) were calculated using two-way ANOVA with repeated measures and the Tukey–Kramer test for multiple comparisons. **(C)** RT-qPCR RVA RNA detection frequency in extraintestinal tissues across dpi 1–4. **(D)** RT-PCR Detection frequency at time points with significant differences. The frequency of RVA detection across tissues from dpi 1–4 was assessed using Fisher's exact test. Significant differences (*$p < 0.05$) are indicated. **(E)** In situ hybridization (ISH)-based detection frequency of RVA RNA in extraintestinal tissues across dpi 1–4.(F) ISH Detection frequency at time points with significant differences. Statistical analysis was performed using Fisher's exact test. Significant differences (*$p < 0.05$) are indicated.

showed a significantly lower frequency of G9P[13] RNA detection (p<0.05) compared to the liver, blood, and salivary glands (Fig 2D).

RNAscope in situ hybridization (ISH) was used to verify the presence of RVA mRNA indicative of actively replicating virus within tissue fragments (see photos, Fig 7C, 7F, 7I, and S1 Picture). RVA ISH positive signals were observed in small intestine, lungs, salivary glands, liver and spleen. Overall, out of 60 RVA-challenged pigs, 31 were ISH-positive in the lungs, 27 in the spleen, 19 in the liver, and 21 in the salivary glands. The analysis of ISH RVA RNA detection frequency (Fig 2E) showed that similar to qRT-PCR data, the most prominent differences between tissues were observed at dpi 4 (Figs 2E, 2F and S2C): liver had a significantly lower frequency of RVA RNA detection compared to the lungs (p<0.05) following G5P[7] infection at dpi 4 (Fig 2F).

Interestingly, despite the substantial differences in intestinal replication between G5P[7] and the other RVA strains used in this study (S1 Fig), the analysis of levels of RVA RNA in extraintestinal tissues did not reveal significant differences between RVA strains (Figs 3,4 and S3). While numerical differences in RVA RNA levels in the lungs and liver across dpi 1–4 (Fig 3) were observed, this difference did not reach statistical significance, primarily due to the high standard deviation. The analysis of the proportion of positive samples across dpi 1–4 (Fig 4A and 4B) revealed that G9P[13] had a significantly higher frequency of RT-qPCR RVA RNA detection in salivary glands compared to G5P[7] (p<0.05) at dpi 3–4 (Fig 4B).

The strain-specific analysis of ISH RVA RNA detection frequency (Figs 3D, 3E, S4 and S5) demonstrated that across dpi1–4 tissues from GF pigs infected with G9P[13] had significantly higher number of RVA RNA frequency in liver at dpi 1 and 4.

## The limited availability of RVA receptors in extraintestinal tissues may restrict its replication

The lack of a strong association between RVA infection and pathological changes in extraintestinal tissues, combined with significantly lower titers compared to its replication levels in the intestine, suggests the presence of factors that limit extraintestinal replication of RVA. We analyzed the distribution of a variety of receptors involved in RVA attachment and binding, including SA-containing and SA-non-containing glycans (Fig 5).

The expression levels varied significantly among the tissues (Fig 5B). The levels of H-antigen (detected by BG4 antibody) expression in the ileum and lungs were significantly (ileum) or marginally (lungs) higher than those in the liver, spleen, and salivary glands. All tissues exhibited relatively high levels of 2,6 SA (detected by SNA lectin), with no significant difference among the tissues. However, only intestinal tissues showed expression of 2,3 SA (detected by MAL-1 lectin). Non-sialylated galactose (detected by PNA lectin) was uniquely expressed at the highest level in ileum while the rest of the tissues did not show considerable levels of expression of this type of glycan. The expression of glycans terminating in α-galactose (GSL-1 lectin) was prominent in ileum and lungs compared to the rest of the tissues. These findings indicate the highest overall diversity and abundance of RVA receptors in the gut, followed by the lungs.

## Rotavirus A induces gross and microscopic lesions in the lungs

During the necropsies, pleural and pulmonary multifocal to coalescing, mild to moderate areas of hemorrhage, along with multifocal mild atelectasis were observed predominantly in the dorsal surface of caudal lobes and less frequently in the cranial lobes of pigs challenged with RVA strains but were not present in pigs from the negative control group (Fig 6). Gross lesions were not observed in other organs from challenged nor negative control pigs. Histological lesions were assessed on H&E-stained tissue fragments, with lesions observed only in lungs and small intestine (Fig 7B, 7E and S1 Data) but not in other organs including salivary glands (Fig 7G-H). In lungs, focal to multifocal moderate intralveolar and subpleural hemorrhagic areas and mild multifocal areas of atelectasis were also observed in fragments from RVA challenged pigs (Fig 7E), but not in pigs from the negative control (Fig 7D). In small intestine, vacuolar degeneration of

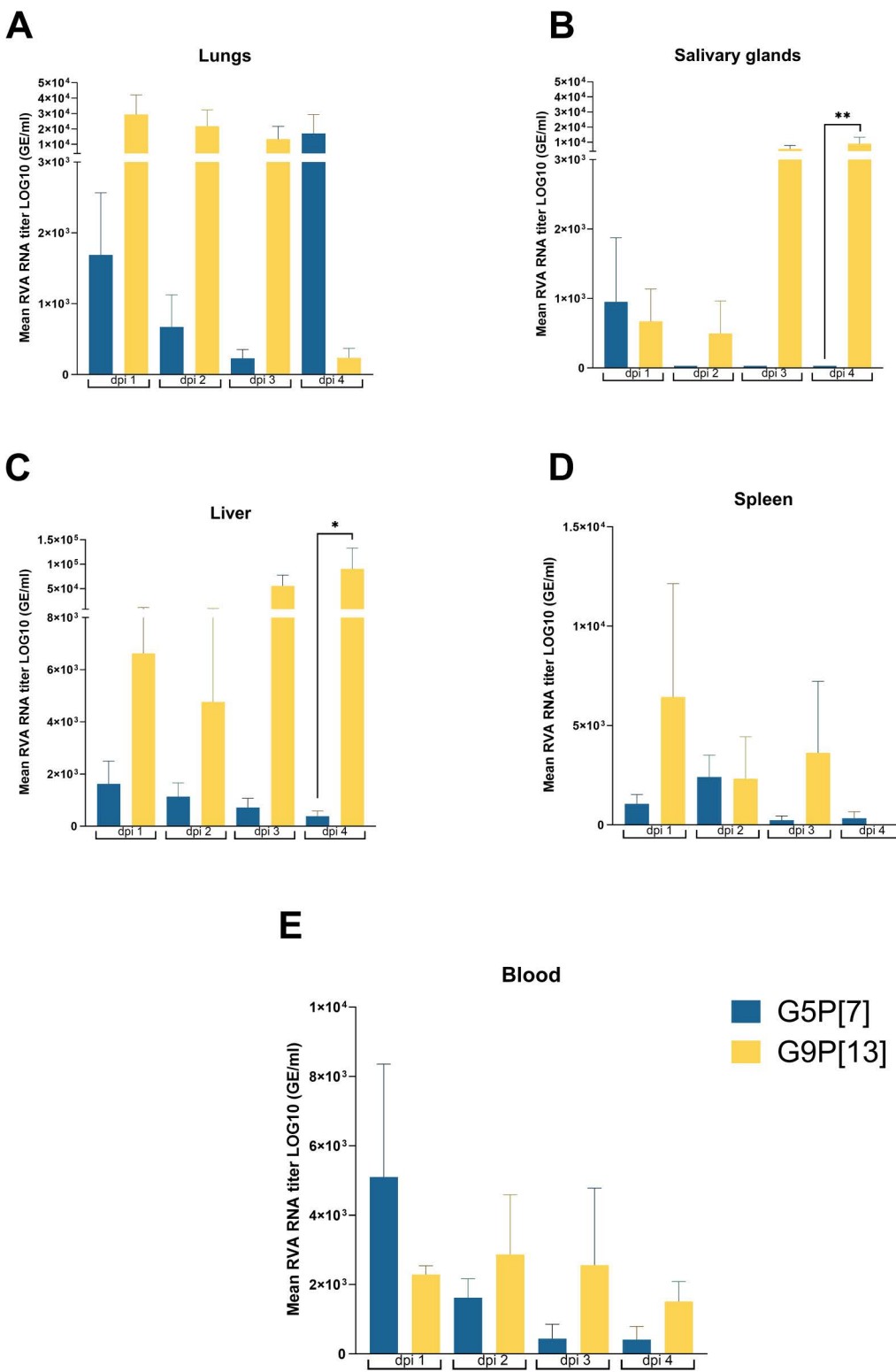

**Fig 3. Strain-specific analysis of RVA RNA levels in extraintestinal tissues.** Six-day-old germ-free pigs were orally inoculated with $1 \times 10^6$ FFU of each RVA. At the indicated post-inoculation time-points, animals were euthanized, and tissues (A: lungs; B: salivary glands; C: liver; D: spleen and E: blood) were collected. Significant differences (*p < 0.05, **p < 0.01) were determined using two-way ANOVA with repeated measures and the Tukey–Kramer test for multiple comparisons.

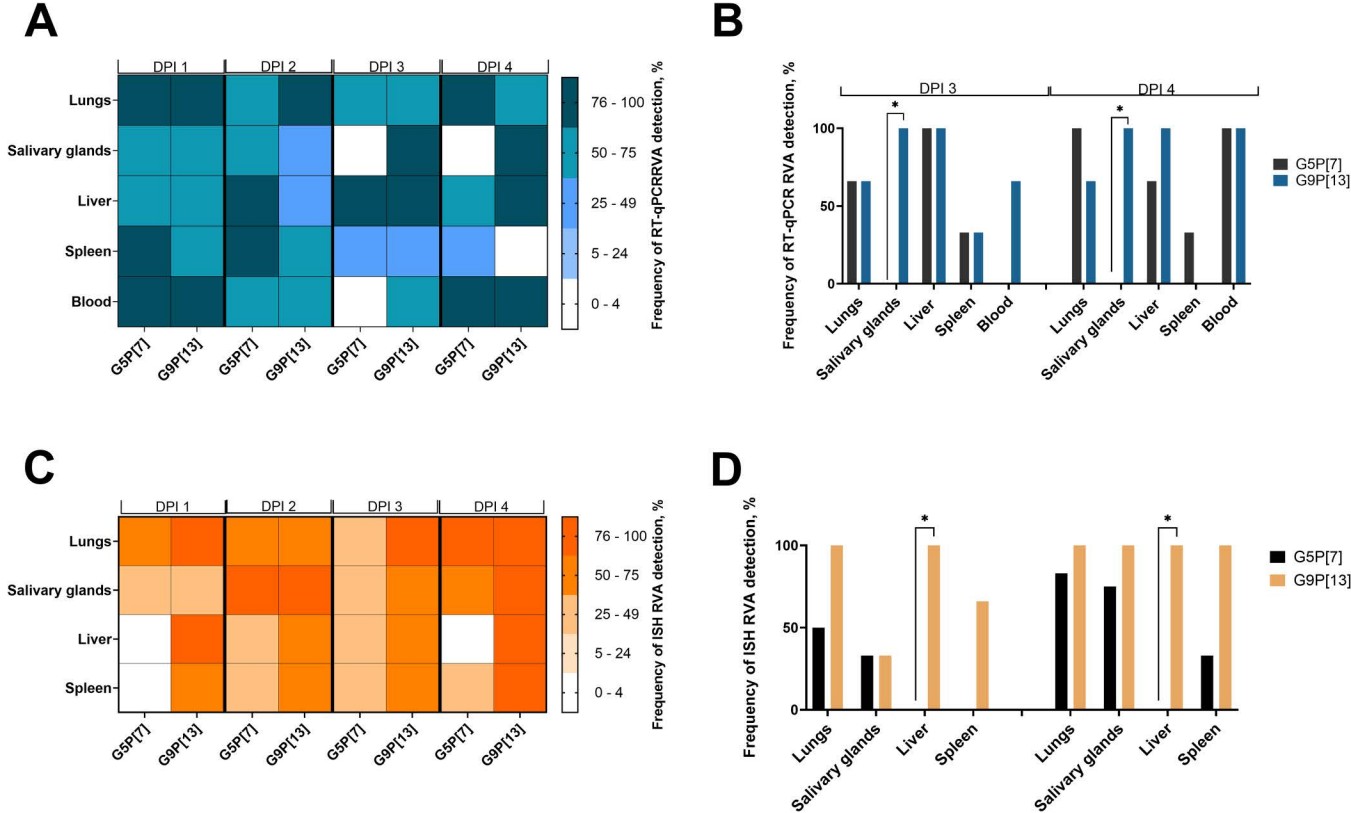

**Fig 4. Strain-specific analysis of RVA RNA levels in extraintestinal tissues.** Six-day-old germ-free pigs were orally inoculated with $1 \times 10^6$ FFU of each RVA. At the indicated post-inoculation time points, animals were euthanized, and tissues (lungs, salivary glands, liver, blood, and spleen) were collected. **(A)** RT-qPCR RVA RNA detection frequency across dpi 1–4; **(B)** Detection frequency at time points where significant differences were observed. The frequency of RVA detection across tissues from dpi 1–4 was assessed using Fisher's exact test. Significant differences (*$p < 0.05$) are indicated. **(C)** ISH-based RVA RNA detection frequency across dpi 1–4; **(D)** Detection frequency at time points with significant differences. Statistical analysis was performed using Fisher's exact test. Significant differences (*$p < 0.05$) are indicated.

enterocytes in the tip of villi and villus atrophy and blunting were observed in small intestine (Fig 7B) of some RVA challenged pigs but not in pigs from the negative control (Fig 7A).

## Discussion

RVA is ubiquitous, and despite vaccine availability it remains a leading cause of diarrhea in children under five years of age. Previous studies have indicated the potential of RVA to induce pathology outside the intestine [4–6,9,23]. In this study, we aimed to determine whether increased intestinal replication of RVA would lead to its increased dissemination to extraintestinal tissues. Among all RVA strains tested in this study, G9P[13] but not G5P[7], exhibited a higher frequency of spread to extraintestinal tissues despite the relatively moderate levels of intestinal replication observed for G9P[13]. These findings suggest that the systemic dissemination of RVA may be influenced by factors beyond the degree of its intestinal replication.

Our data demonstrated that all RVA strains used in this study were able to spread outside the intestine: RVA RNA was detected in blood, lungs, salivary glands, liver, and spleen with the lungs exhibiting the highest frequency of RVA RNA detection. However, consistent with previously published data [9,10], the levels of RVA RNA in these tissues were low. In addition, while a typical biphasic RVA intestinal shedding pattern was observed, the detection of RVA RNA in

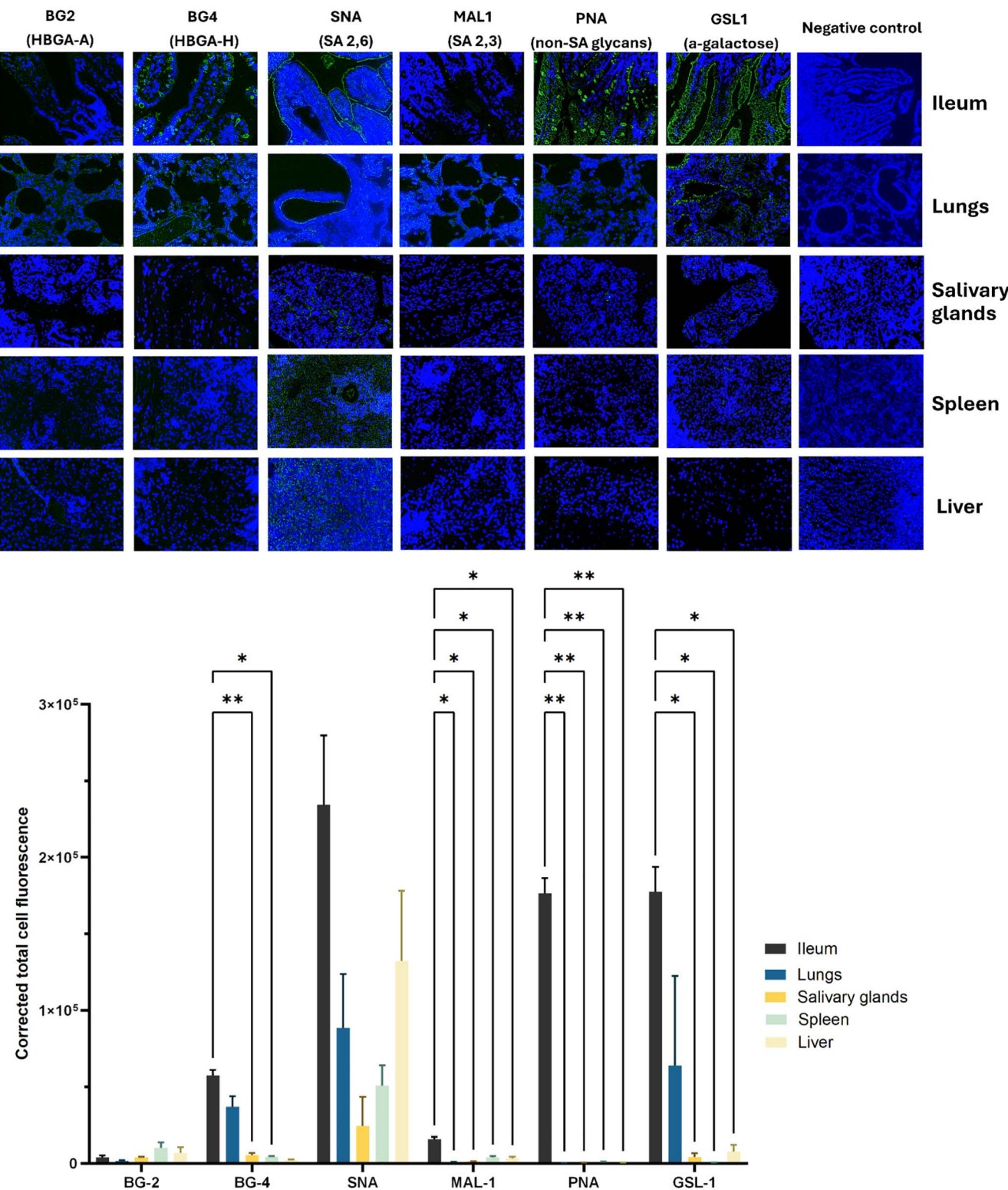

**Fig 5. Distribution of sialylated and non-sialylated glycans across the intestinal and extraintestinal tissues.** Representative fluorescence microscopy images (A) show DAPI-stained nuclei (blue), highlighting tissue architecture. Sialylated glycans were detected using FITC-labeled Sambucus nigra agglutinin (SNA) and Maackia amurensis I lectin (MAL **I**), which are specific to α2,6- and α2,3-linked sialic acids, respectively. Non-sialylated glycans were detected using FITC-labeled peanut agglutinin (PNA) and Griffonia simplicifolia lectin I (GSL **I**), which are specific to galactose and α-galactose,

respectively (green).Histo-blood group antigens A (HBGA-A) and H (HBGA-H) were detected using specific antibodies, followed by FITC-labeled goat anti-mouse IgG/A/M (H+L) secondary antibody (1:750; Bio-Rad).The expression levels of glycans were analyzed (B) by calculating the corrected total cell fluorescence (CTCF) [47].

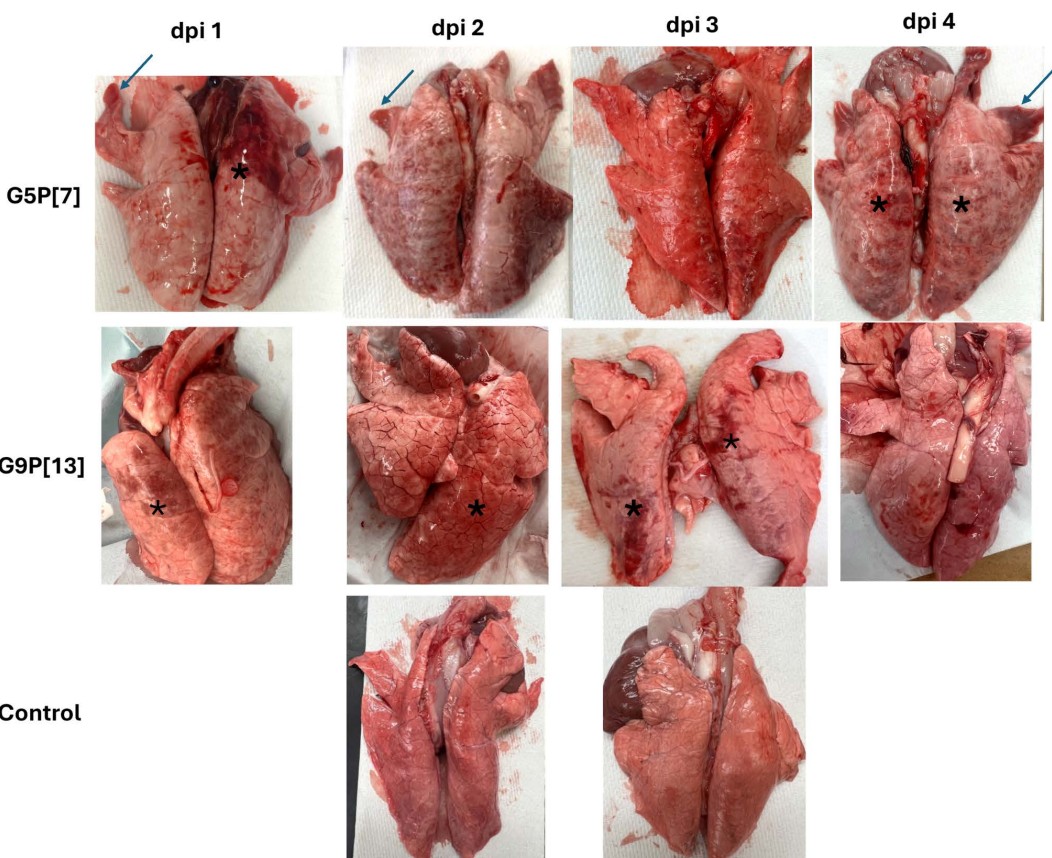

**Fig 6. Gross evaluation of extraintestinal organs detected changes in lungs from germ-free pigs challenged with RVA.** Lungs from negative control group pigs were fully collapsed and did not present any gross changes (left), while lungs from Rotavirus A–challenged pigs did not collapse and showed areas of red discoloration (black asterisks) and atelectasis (blue arrows).

extraintestinal organs occurred sporadically over the experimental period (dpi 1–4). These findings suggest that RVA can transiently disseminate beyond the gut, highlighting the importance of extraintestinal involvement in RVA pathogenesis, including early stages of infection. This also underscores the need for studies focusing on the very early phase of infection (within the first 24 hours post-inoculation), to better understand the timing and mechanisms of viral dissemination.

Previous attempts have been made to identify the role of virus phenotype as a key factor affecting enteric virus escape to extraintestinal tissues. The sialic acid dependent reovirus [24] and simian rhesus RVA [9] were found to replicate in extraintestinal tissues more efficiently compared to sialic acid independent strains. Recent studies indicated that certain RVs including G9P[13] replicated at higher levels in vitro after sialic acid removal [13,20,21] while the replication of G5P[7] was significantly inhibited [13]. The current study demonstrated that the extraintestinal spread of the sialic acid-dependent

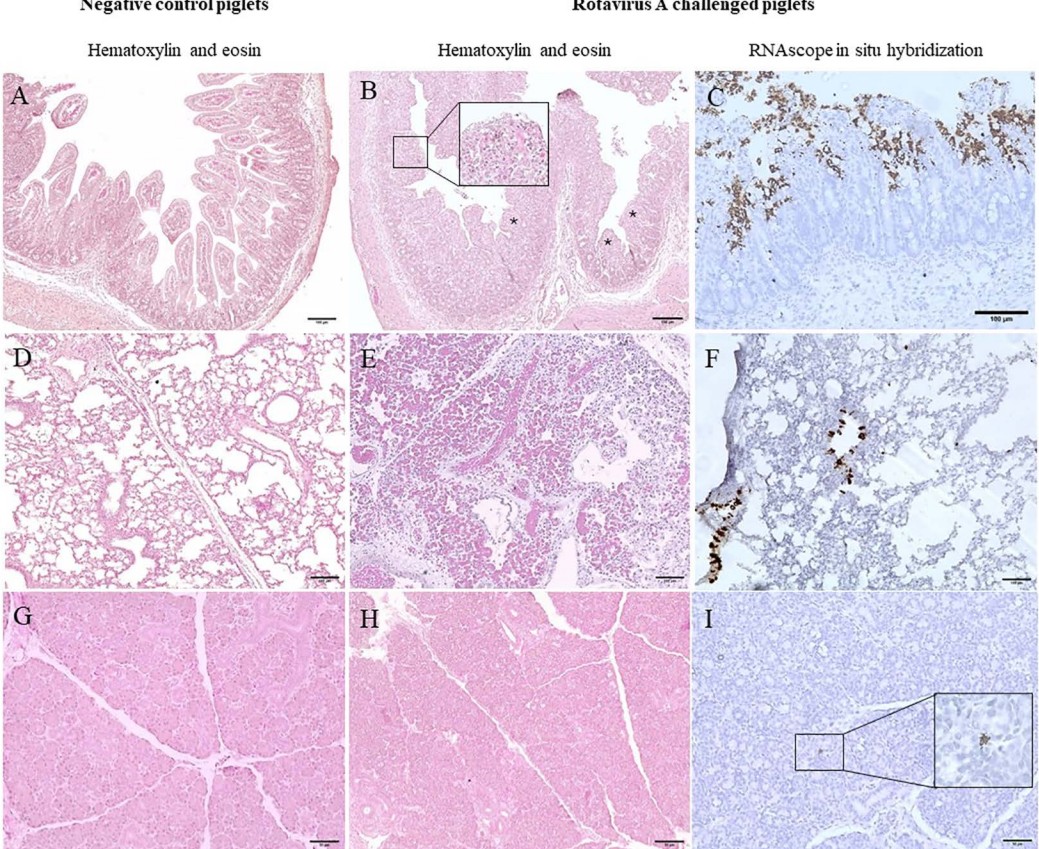

**Fig 7. Histological changes and positive in situ hybridization signals were observed in germ-free pigs experimentally challenged with RVA.** **(A)** Ileum from a negative control piglet without histological changes. **(B)** Ileum from a Rotavirus A (G5P[7]) challenged piglet with villus atrophy and blunting (asterisks and degeneration of epithelial cells in the tip of the villi (insert). **(C)** Ileum villus epitheilal cells with multifocal positive signals (brown) for Rotavirus A (G5P[7]). **(D)** Lung from a negative control group piglet without histologic changes. **(E)** Lung of a Rotavirus A (G5P[7]) challenged piglet with severe hyperemia. **(F)** Lung of a Rotavirus A (G5P[7]) challenged piglet showing in situ hybridization positive signals (brown) in epithelial cells of bronchioles. **(G)** Salivary gland from a negative control piglet without histologic changes. **(H)** Salivary gland from a Rotavirus A (G5P[7]) challenged piglet without histological changes. **(I)** Salivary glands with focal positive in situ hybridization signal (brown) for Rotavirus A (G5P[7]).

(=sialidase sensitive) strain G5P[7] was even less efficient than that of G9P[13], suggesting that other viral determinants or host factors, such as the diversity of RVA receptors, could contribute to RV dissemination outside the intestine.

The RV-host interactions involve multiple receptors, including a wide spectrum of sialic acid-containing and non-sialic acid-containing molecules – glycans whose diversity significantly influences RVA replication [3]. In this study, while several RVA receptors were expressed in extraintestinal tissues, intestinal tissues exhibited the highest glycan diversity. For example, only intestinal tissues demonstrated considerable levels of α2,3 sialic acid expression. While only limited data is available on RVA α2,3/2,6 preferences [18], our data may indicate that presence of α2,6 sialic acid may be a key factor for RVA replication. Among the extraintestinal tissues, lungs were the most glycan-diverse tissue, uniquely expressing considerable levels of HBGA and terminal α-galactose, which coincided with the highest proportion of RVA RNA-positive samples among extraintestinal tissues. Lungs have been previously identified as one of the primary sites of extraintestinal replication [23,25,26]. In addition, recent studies demonstrated a strong association between RVA and pathomorphological changes in lungs [7,22]. By using germ-free pigs, we have demonstrated clear evidence of an association between the presence of RVA in the lungs and the development of both gross and microscopic lesions. Interestingly, while we observed

pathological changes in the lungs, the level of viral replication in this tissue was very low. A recent study [27] demonstrated that rotavirus-induced diarrhea can occur in the absence of active viral replication, suggesting that innate immune activation by non-infectious viral particles may contribute to tissue pathology.

Interestingly, the salivary glands - the tissue which exhibited the least prominent expression of α2,3- and α2,6-linked sialic acids, showed a significantly higher proportion of RVA RNA-positive samples following G9P[13] compared to G5P[7] infection, further emphasizing the G9P[13] preference for non-sialylated glycans [13]. While there were no macro- or microscopic changes in the salivary glands, liver and spleen, the presence of RVA outside the intestine remains significant beyond its potential role in pathology. Replication of RVA in extraintestinal tissues, such as salivary glands and respiratory tract, is significant as it may contribute to non-enteric routes of viral transmission [6,28,29]. In addition, RVA presence in hepatic and spleen tissue could suggest a potential role of these organs in systemic dissemination of RVA. This highlights a potential association between receptor diversity and RVA replication and indicates that the frequent detection of G9P[13] in extraintestinal tissues, particularly in the lungs and salivary glands, suggests a potential mechanism for this virus strain's global spread [30–32]. In addition to the presence of RVA receptors, other host factors—such as presence or levels of trypsin-like serine proteases [33]—may regulate viral replication and tissue tropism. These proteases, abundant in the porcine intestine and required for RVA VP4 cleavage, are also present at lower levels in tissues like the lung, salivary glands, and liver [34,35]. Other classes of proteases, including threonine hydrolases [36], have also been implicated in viral protein processing in certain tissues, though their specific role in RVA activation remains to be explored. Thus, limited extraintestinal replication may result not only from receptor availability but also from the local abundance and diversity of activating proteases, highlighting additional mechanisms that could explain the observed viral spread.

In conclusion, our study provides the first compelling evidence that RVA RNA levels in extraintestinal tissues are independent of high intestinal viral loads. The lungs emerged as a key site of RVA extraintestinal spread, correlating with unique glycan expression profiles and the associated pathological changes. Differences in glycan receptor diversity, particularly the presence of non-sialylated structures and HBGAs, may underline the broader tissue tropism observed for strains like G9P[13]. These findings emphasize the importance of both viral and host factors in shaping RVA dissemination and highlight the potential for non-enteric transmission routes.

## Materials and methods

### Ethics statement

All animal experiments were approved by the Institutional Animal Care and Use Committee (IACUC) at The Ohio State University).

### Rotaviruses

Virulent strains of RVA: human Wa G1P[8] [37] and porcineRV0084 G9P[13] [38], Gottfried G4P[6], and OSU G5P[7] [39] maintained by serial passage in germ-free (GF) pigs, were used to orally inoculate pigs.

### Animal experiments

Near-term pregnant commercial sows (Landrace × Yorkshire × Duroc crossbred) were purchased from The Ohio State University swine facility or from Shoup Brothers Farm LTD, Ohio, USA and housed in OSU animal facilities. Pigs were derived by cesarean section and maintained as described previously [40]. Rectal swabs were taken from all the pigs at two days of age, and sterility was confirmed by culturing of rectal swabs in blood agar plates and thioglycolate broth culture. Six- to eight-day-old GF pigs were inoculated with individual RVA strains (n = 12–24 per strain, S1 Table) at a dose of $1 \times 10^6$ FFU/piglet (Fig 8). At 1–4 days post-inoculation (dpi) pigs (n = 3–6) were euthanized, and rectal swabs, blood, intestinal contents, and tissue samples (lungs, salivary glands, ileum, jejunum, spleen, liver, pancreas) were collected. Rectal swabs

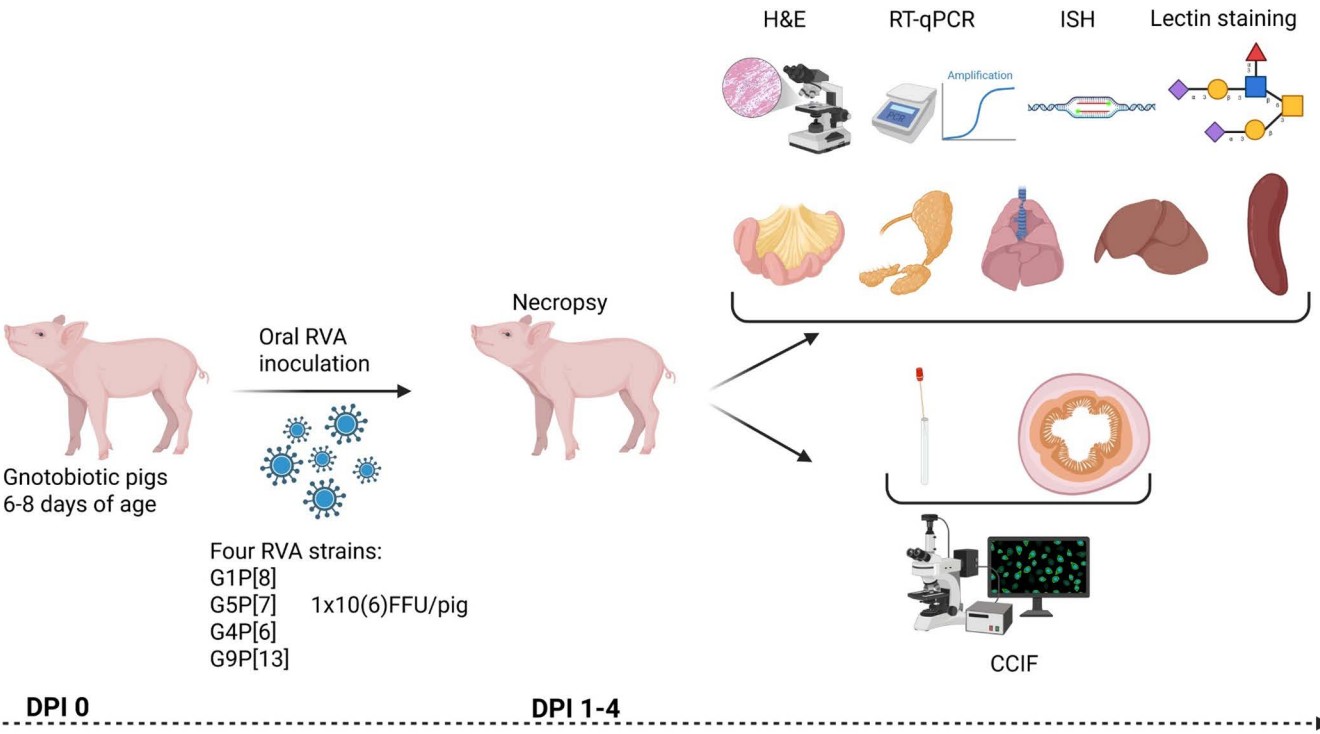

**Fig 8. The scheme illustrates experimental design.** Six- to eight-day-old germ-free pigs were inoculated with an individual RVA strain at a dose of $1 \times 10^6$ FFU/piglet. At 1–4 days post-inoculation (dpi), pigs (n = 3–6) were euthanized, and rectal swabs, blood, intestinal contents, and tissue samples (lungs, salivary glands, ileum, jejunum, spleen, liver, pancreas) were collected. Rectal swabs were used to assess RV shedding by cell culture immunofluorescence. Tissue fragments were either fixed in 10% neutral buffered formalin, embedded in paraffin, and stored at room temperature for future RNA in situ hybridization and hematoxylin and eosin staining or frozen at −20°C for future RNA extraction and RT-qPCR. Created in BioRender. Raev, **S.** (2025). https://BioRender.com/ctpuk1b.

were used to assess RV shedding by cell culture immunofluorescence (CCIF). Tissue fragments were either fixed in 10% neutral buffered formalin, embedded in paraffin, and stored at room temperature for future RNA in situ hybridization (ISH) and H&E staining or frozen at -20 °C for future homogenization, RNA extraction using MagMAX Viral/Pathogen Nucleic Acid Isolation Kit (Thermo Fisher Scientific). The 87 bp RVA-specific NSP3 sequence (S2 Table) was amplified and cloned using the TOPO TA kit (Thermo Fisher Scientific, MA, USA). Linearized plasmid was quantified by NanoDrop® 2000 and serially diluted ($1 \times 10^{10}$ to 0.25 copies/µL) with three replicates per dilution to generate a standard curve. Identical primers were used for all RVA strains. RNA from RVB, RVC, and uninfected piglets or PIEs served as negative controls. The standard curve had a slope of −3.053 (PCR efficiency 113%) and R2 = 0.998, confirming highly linear amplification. Levels of RVA RNA were determined by RT-qPCR and expressed as $\log_{10}$ GE/mL by using one step RT-qPCR Qiagen kit (QIAGEN, Germantown, MA, USA) as described before [41,42]. During the necropsies, tissues were also grossly evaluated.

### RNA in situ hybridization

ISH was performed according to the manufacturer's instructions using RNAscope (Advanced Cell Diagnostics). An RVA probe targeting the VP6 gene [43], as well as host (*Sus scrofa*)-specific and non-specific probes served as positive and negative controls, respectively, ensuring the specificity of the assay. Additionally, tissues from non-challenged GF pigs served as an additional negative control.

## Rotavirus fecal shedding

CCIF assay was used to quantify RVA in rectal swabs and intestinal contents as previously described [44,45]. Briefly, serial 10-fold dilutions of processed samples in MEM containing 1 µg/mL porcine trypsin (Sigma-Aldrich, USA) were added to confluent monolayers of MA-104 cells in 96-well plates, centrifuged at 1,200 × g for 30 min, and incubated at 37°C for 18 hours. After fixation of cells with 80% acetone, monoclonal RVA-VP6-specific antibodies (RV RG23B9C5H11) diluted 1:100 in PBS were added and incubated overnight at 4°C. Following PBS washes, secondary FITC-conjugated goat anti-mouse antibodies (SeraCare Life Sciences, USA) were applied. The final RVA titers were calculated and expressed as the reciprocal of the highest dilution at which positive fluorescing cells were observed.

## Immunofluorescence for sialylated and non-sialylated glycans

The presence and distribution of sialylated and non-sialylated glycans was assessed by immunofluorescence using antibodies/lectins listed in Table 1.

Formalin-fixed-paraffin-embedded sections were baked at 60°C for 1 hour, treated with xylene for 20 minutes, washed with 100% ethanol for 20 minutes, then incubated with 90, 70 and 50% ethanol for 5 minutes each. The deparaffinized sections were boiled in citrate buffer (pH 6.0) at 95°C for 15 minutes for antigen retrieval, washed with 0.05 M Tris Buffer Saline (TBS) (pH 7.6) containing 0.05% Tween 20 (TBST) and then incubated with 0.5% Triton X-100 for 10 minutes. The sections were washed 3 times with TBST, and then either individual lectins or mAbs (Table 1) were added and incubated for 1 hour at room temperature in the dark. For HBGA detection, slides were washed 3 times with TBST and secondary antibodies (FITC labeled goat anti-mouse IgG/A/M (H/L), 1:750, BioRad) were applied. All secondary antibodies were incubated with the slides at room temperature for 1hour. DAPI (4′,6-diamidino-2-phenylindole) was used for nuclear staining and the Vector TrueVIEW Autofluorescence Quenching Kit was used to reduce background fluorescence. Slides were sealed by adding mounting medium and pictures were taken using Keyence BZ-810 microscope. Lactose at 400mM concentration was used as a haptenic sugar to prove lectin binding specificity [46]. The expression levels of glycans were analyzed by calculating the corrected total cell fluorescence (CTCF) using the formula: CTCF = integrated density – (area of the selected cell × mean fluorescence of background readings), as described previously [47].

## Histopathology

Formalin-fixed, paraffin-embedded tissue sections were stained with hematoxylin and eosin (H&E) using standard protocols [48]. Lesions were assessed for severity, distribution, and morphological features based on established criteria.

## Statistical analysis

All statistical analyses were performed using GraphPad Prism version 10 (GraphPad Software, Inc., La Jolla, CA, USA). Fecal RV shedding and RVA replication levels in extraintestinal tissues post-challenge were analyzed using two-way

**Table 1. Lectins and antibodies used in this study.**

| Target | Reagent | Working concentration/dilution | Source |
|---|---|---|---|
| Lectins | | | |
| α2,3 sialic acid | Maackia amurensis I lectin (MAL I) | 5µg/ml | Vector Laboratories |
| α2,6 sialic acid | Sambucus nigra lectin (SNA) | | |
| Non-sialylated α-linked galactose | Peanut agglutinin lectin (PNA) | | |
| Terminal α-galactose | Griffonia Simplicifolia Lectin I (GSL I) | | |
| Antibodies | | | |
| Histo-blood group antigen(HBGA)-A | BG-2, anti-A antibodies | 1:50 | Biolegend |
| Histo-blood group antigen-H | BG-4, anti-H type 1 antibodies | | |

ANOVA with repeated measures, followed by the Tukey-Kramer test for multiple comparisons. The frequency of RVA detection across extraintestinal tissues from dpi 1–4 was assessed using Fisher's exact test. Differences were considered significant at $p \leq 0.05$.

## Supporting information

**S1 Fig. RVA levels in the small intestine (A) and large intestine (B); RVA shedding (rectal swabs) (C).** Six-day-old germ-free pigs were orally inoculated with $1 \times 106$ FFU of rotavirus. Swabs were collected at designated time points. For the small and large intestine contents pigs were euthanized at the post-inoculation times indicated and contents were collected. RVA quantification was performed with cell culture immunofluorescence (CCIF). Significant differences (*p < 0.05, ** p < 0.01, p < 0.001) are indicated as calculated by using two-way ANOVA with repeated measures and the Tukey-Kramer test for multiple comparisons.
(TIF)

**S2 Fig. Tissue-specific analysis of RVA RNA levels in extraintestinal tissues.** Six-day-old germ-free pigs were orally inoculated with $1 \times 106$ FFU of rotavirus. At the post inoculation times indicated, animals were euthanized and tissues (lungs, salivary glands, liver, blood and spleen) were collected. (A) Levels of RVA RNA in extraintestinal tissues across dpi 1–4. (B) RT-qPCR RVA RNA detection frequency in extraintestinal tissues across dpi 1–4. (C) ISH RT-qPCR RVA RNA detection frequency in extraintestinal tissues across dpi 1–4 and (D) at time points when significant differences were observed. Statistical analysis for RVA RNA detection frequency was performed by using Fisher's exact test. Significant differences (*p < 0.05) are indicated.
(TIF)

**S3 Fig. Strain-specific analysis of RVA RNA levels in extraintestinal tissues.** Six-day-old germ-free pigs were orally inoculated with $1 \times 10^6$ FFU of each rotavirus. At the indicated post-inoculation time points, animals were euthanized, and tissues (A: lungs; B: salivary glands; C: liver; D: spleen and E: blood) were collected. Significant differences (*p < 0.05, **p < 0.01) were determined using two-way ANOVA with repeated measures and the Tukey–Kramer test for multiple comparisons.
(TIF)

**S4 Fig. Strain-specific analysis of RVA RNA levels in extraintestinal tissues.** Six-day-old germ-free pigs were orally inoculated with $1 \times 10^6$ FFU of each rotavirus. At the indicated post-inoculation time points, animals were euthanized, and tissues (lungs, salivary glands, liver, blood, and spleen) were collected. (A) RT-qPCR RVA RNA detection frequency across dpi 1–4; (B) Detection frequency at time points where significant differences were observed. The frequency of RVA detection across tissues from dpi 1–4 was assessed using Fisher's exact test. Significant differences (*p < 0.05) are indicated. (C) ISH-based RVA RNA detection frequency across dpi 1–4; (D) Detection frequency at time points with significant differences. Statistical analysis was performed using Fisher's exact test. Significant differences (*p < 0.05) are indicated.
(TIF)

**S5 Fig. Strain-specific analysis of RVA levels (CCIF) in extraintestinal tissues.** Six-day-old germ-free pigs were orally inoculated with $1 \times 10^6$ FFU of each rotavirus. At the indicated post-inoculation time points, animals were euthanized, and tissues (A: lungs; B: salivary glands; C: liver; D: spleen and E: blood) were collected. Significant differences (*p < 0.05, **p < 0.01) were determined using two-way ANOVA with repeated measures and the Tukey–Kramer test for multiple comparisons.
(TIF)

**S1 Table. Number of Gnotobiotic Piglets Used in This Study by Rotavirus A Strain and Days Post-Infection.**
(DOCX)

**S2 Table. Primers and probes used in this study.**
(DOCX)

**S1 Picture. Spleen (A) and liver (B) with focal positive in situ hybridization signal (brown) for Rotavirus A (G5P[7]).**
(TIF)

**S1 Data. All Data Supporting the Findings of This Study.**
(XLSX)

## Acknowledgments

We thank Ronna Wood, Joshua Amimo and Juliette Hanson for their technical assistance.

## Author contributions

**Conceptualization:** Sergei A Raev, Anastasia N. Vlasova.

**Data curation:** Sergei A Raev.

**Formal analysis:** Sergei A Raev, Talita P Resende.

**Funding acquisition:** Anastasia N. Vlasova.

**Investigation:** Sergei A Raev, Maryssa K Kick, Maria Chellis.

**Methodology:** Sergei A Raev, Maryssa K Kick, Maria Chellis, Talita P Resende.

**Project administration:** Anastasia N. Vlasova.

**Resources:** Anastasia N. Vlasova.

**Supervision:** Anastasia N. Vlasova.

**Visualization:** Sergei A Raev.

**Writing – original draft:** Sergei A Raev.

**Writing – review & editing:** Sergei A Raev, Linda J Saif, Anastasia N. Vlasova.

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
