## [Decision Letter · Decision Letter 0]

25 Aug 2025

The efficiency of rotavirus A spread to extraintestinal tissues is not determined by the levels of its replication in the gut

PLOS Pathogens

Dear Dr. Vlasova,

Thank you for submitting your manuscript to PLOS Pathogens. After careful consideration, we feel that it has merit but does not fully meet PLOS Pathogens's publication criteria as it currently stands. Therefore, we invite you to submit a revised version of the manuscript that addresses the points raised during the review process.

Please submit your revised manuscript within 60 days Oct 24 2025 11:59PM. If you will need more time than this to complete your revisions, please reply to this message or contact the journal office at plospathogens@plos.org. Please include the following items when submitting your revised manuscript:

We look forward to receiving your revised manuscript.

Kind regards,

Karl Boehme, PhD

Guest Editor

PLOS Pathogens

Alexander Gorbalenya

Section Editor

PLOS Pathogens

Editor-in-Chief

PLOS Pathogens

orcid.org/0000-0003-2946-9497

Michael Malim

PLOS Pathogens

orcid.org/0000-0002-7699-2064

**Journal Requirements:**

At this stage, the following Authors/Authors require contributions: Sergei A Raev, Maryssa K Kick, Maria Chellis, Linda J Saif, Talita P Resende, and Anastasia Nickolaevna Vlasova. Please ensure that the full contributions of each author are acknowledged in the "Add/Edit/Remove Authors" section of our submission form.

- TM on page: 15.

5) We notice that your supplementary Figures, and Tables are included in the manuscript file. Please remove them and upload them with the file type 'Supporting Information'. Please ensure that each Supporting Information file has a legend listed in the manuscript after the references list.

Potential Copyright Issues:

i) Please confirm (a) that you are the photographer of 5, or (b) provide written permission from the photographer to publish the photo(s) under our CC BY 4.0 license.

ii) Please confirm (a) that you are the photographer of 5, or (b) provide written permission from the photographer to publish the photo(s) under our CC BY 4.0 license.

7)  We note that figure 7 is created through BioRender. Please confirm that you hold a Premium account and provide a pdf copy of the CC BY 4.0 Licence as provided by BioRender. For instructions on how to generate a CC BY 4.0 license for your figure, please see the guidelines here: https://help.biorender.com/hc/en-gb/articles/21282341238045-Publishing-in-open-access-resources. 

If you are using the free assets from BioRender, we are unable to publish these images as they are licenced under a stricter licence than CC BY 4.0. In this case we ask you to remove the BioRender images and replace them with open source alternatives.

See these open source resources you may use to replace images / clip-art:

- https://bioart.niaid.nih.gov/

- https://bioicons.com/

- https://healthicons.org/

- https://scidraw.io/

- https://reactome.org/icon-lib

- https://www.phylopic.org/images

- https://journals.plos.org/plosbiology/article?id=10.1371/journal.pbio.3002395

8) We note that your Data Availability Statement is currently as follows: "All relevant data are within the manuscript.". Please confirm at this time whether or not your submission contains all raw data required to replicate the results of your study. Authors must share the “minimal data set” for their submission. PLOS defines the minimal data set to consist of the data required to replicate all study findings reported in the article, as well as related metadata and methods (https://journals.plos.org/plosone/s/data-availability#loc-minimal-data-set-definition).

9) Please amend your detailed Financial Disclosure statement. This is published with the article. It must therefore be completed in full sentences and contain the exact wording you wish to be published.

10) Kindly revise your competing statement to align with the journal's style guidelines: 'The authors declare that there are no competing interests.'

**Reviewers' Comments:**

Reviewer's Responses to Questions

**Part I - Summary**

Reviewer #1: This paper shows that RVA's extraintestinal spread does not depend on its replication efficiency in the gut, which may explain the current global dominance of G9P[13] and related RVA strains. In particular, the authors show that the G9P[3] strain could be capable of airborne spread, as the lungs have the highest frequency of RVA RNA detection, the highest host glycan diversity/abundance, and the presence of lesions both macroscopically and histologically. The authors' claims are substantiated by their well-structured experiments and research results, making this paper worthy of publication in PLoS Pathogens.

Reviewer #2: This manuscript investigates the extraintestinal dissemination of Rotavirus A (RVA) in gnotobiotic pigs, focusing on correlation between viral replication in the gut and its spread to other organs. The authors utilized four RVA strains with differing replication capacities—human Wa (G1P[8]), porcine RV0084 (G9P[13]), Gottfried (G4P[6]), and OSU (G5P[7]). Remarkably, the G9P[13] strain, which replicated to a lesser extent in the intestine, exhibited more efficient dissemination to extraintestinal tissues. These findings suggest that RVA extraintestinal spread is not determined by intestinal replication efficiency and that viral genomes detected in peripheral organs are not merely residual byproducts of intestinal replication. Furthermore, the authors demonstrate clear macroscopic and microscopic lesions in the lungs, suggesting that RVA infection may contribute to respiratory pathology—consistent with earlier observations in rodent models.

However, several limitations in the dataset make it difficult to draw definitive conclusions regarding true extraintestinal replication. In particular, the heatmap-style data presentation obscures the time-course dynamics of viral genome levels in each organ—information that is critical to distinguishing between residual viral RNA and active replication. Additionally, no data on the presence of infectious virus (e.g., focus-forming units) in extraintestinal tissues are provided, making it difficult to assess whether productive infection occurred.

While extraintestinal dissemination of RVA has been reported in neonatal mouse, rat, and piglet models (as cited in the manuscript), this study is unique in demonstrating consistent pathological changes in the lungs of infected piglets. Nonetheless, further evidence is needed to confirm that these changes are due to active viral replication and to clarify the functional significance of the observed pathology.

Reviewer #3: In the current study, Raev et al. orally inoculated gnotobiotic piglets with three porcine (G5P[7], G9P[13], and G4P[6]) and one human (G1P[8]) species A rotavirus (RVA). They hypothesized that RVAs that replicate more efficiently in the gut eould spread more efficiently to extraintestinal sites. They characterized viral replication, the presence of viral RNA, and signs of pathology in the intestine and at sites of dissemination, with a focus on two of the viruses (G5 and G9). Rectal virus shedding was higher for G5 than G9 (significant at some time points). However, RTqPCR and RNAish suggested a trend towards more frequent and/or higher levels of extraintestinal dissemination, particularly to the lungs, liver and salivary glands, for G9 than G5. The authors investigated the presence of several glycans known to serve as receptors for some RVAs in various piglet tissues and found the highest levels and diversity of these molecules present in the ileum followed by the lung. Evidence of gross pathology was detected in the lung, histological changes and viral antigen were detected in the lung and gut, and viral antigen was detected in the liver for at least some of the viruses, although the frequencies for each virus are unclear.

Limited numbers of labs possess the capacity to conduct rotavirus infection studies in gnotobiotic piglets. The study found, unexpectedly, that the efficiency of RVA intestinal replication does not necessarily correlate with the efficiency of extraintestinal spread. This finding is significant and interesting and seems to be generally supported by the results. However, by more transparently displaying data points and subsets, the authors might improve confidence in this conclusion. Detection of higher levels of several sialylated and non-sialylated glycans that are receptors for some RVAs in piglet ileum and lungs is consistent with the reasonable and testable hypothesis that virus attachment is an important driver of the observed spread and pathology outcomes. However, this is a biased approach, and other viral and host factors, such as increased presence or diversity of proteases in the gut and lungs or differing sensitivities of viral activation by such proteases, could also explain the outcomes. Since the authors do not directly test the roles of specific glycans in virus attachment, at minimum, alternative explanations should be discussed. As definitive findings from the study are limited, it is essential that the methods are rigorous, and data are presented clearly. However, in several cases, the methods and figure captions lack details that are necessary or would be helpful for interpreting the findings.

**Part II – Major Issues: Key Experiments Required for Acceptance**

Reviewer #1: No further key experiments are required for acceptance.

Reviewer #2: The current data are insufficient to conclusively demonstrate that RVA replicates in extraintestinal organs. As the authors note in the discussion, viral genome copy numbers in these tissues were much lower than in the intestine. It remains unclear whether these represent true replication or are simply residual from systemic dissemination.

Please consider addressing the following:

- Provide time-course plots of viral genome copy numbers for each organ and virus strain individually (redrawing of Figures 1A and 3A) to assess whether viral RNA increases independently of intestinal replication. Avoid using heatmaps for primary quantitative data.

- Include data on the detection of infectious virus (e.g., focus-forming units) in extraintestinal tissues, particularly the lungs.

Reviewer #3: Figs 2,3, S2, and S3 report viral RNA levels in genome equivalents quantified by RT-qPCR, but details of the method are not reported and could not be found in the cited reference. Were identical primers used to quantify all rotavirus strains? Was there a control for specificity, such as RNA from uninfected pigs or PIEs? How was the standard curve generated to quantify genome equivalents? If this information cannot be provided, the authors might need to conduct limited experiments to validate the method.

In the ‘Minor Issues’ section, several suggestions are made regarding data presentation. These include the addition of individual data points to several graphs and information and panels to gross pathological and histological data figures. It is likely that the authors possess the necessary data to make these modifications and will not need to conduct additional experiments to address the comments. If the requested information is provided and rationalized, it is likely these will remain minor concerns/data presentation modifications.

**Part III – Minor Issues: Editorial and Data Presentation Modifications**

Reviewer #1: This manuscript contains several technical errors. Therefore, it will be published once the following errors are corrected.

1. 1. Figure 1 and Supplementary Figure S1 display RVA levels in the small and large intestines for G5P [7] and G9P [13]. However, the p-values for these genotypes differ between the two figures. It is suggested that Figure 1 be deleted and replaced with Supplementary Figure S1.

2. The first letter of words in a subtitle must be capitalized or lowercase.

3. The authors used multiple GN piglets to examine RVA RNA levels in extraintestinal tissues. However, Figures 2D and 2F lack error bars, which suggests that the results are from only one pig. The authors should clarify this point. Is the statistical method used in Figure 2D the two-way ANOVA used in Figure 2B?

4. Figure 4A: The positive fluorescence signals in each panel are hard to see. Organ or tissue names should be labeled. Additionally, each FITC-labeled lectin or HBGAs should be identified.

5. Figure 4B: Write the full names of each lectin and HBGA in the legend for Figure 4B.

6. Figure 5: You used 3-6 GN piglets in each group. However, you did not specify whether the pigs within these groups showed similar lung lesions. Therefore, compile a table with lung lesion indices for these pigs and include a description of these results in the text. Also, add a scale bar to each figure.

7. In the H&E-stained tissue in Figure 6, hematoxylin staining appears weak. This is probably due to inadequate tissue fixation in neutral formalin or insufficient hematoxylin staining. If possible, replace the image with one that shows stronger hematoxylin staining.

8. Describe the virus strain used in Figure 6.

9. Instead of describing the magnification of each panel in the legend of Figure 6, include a scale bar for each panel.

10. Figure 6C legend: Jejunum showing linear positive signals (brown) along the villi surface for Rotavirus A RNA in situ hybridization.

11. The pig schematic for necropsy in the scheme of Figure 7 should be replaced with a piglet.

12. The same full names (abbreviations) are used repeatedly in Materials and Methods.

13. There are no page numbers or line numbers throughout this paper, which makes it difficult to identify what needs to be corrected clearly. Reference #24 should probably replace the one described below in the Materials and Methods section, "Immunofluorescence for sialylated and non-sialylated glycans" (Lin et al., 2017).

Reviewer #2: - Please assign page and line numbers.

- Throughout the manuscript, the terms GF and Gnotobiotic are used. Please unify the terminology (e.g., consistently using “Gnotobiotic pigs” or defining GF at first mention and using the abbreviation thereafter).

- Please enlarge the font size in the figures. Several axis labels, legends, and annotations are difficult to read. Improving legibility would greatly enhance data interpretation.

- Figure 1: Consider using log-scale on the y-axis in panels A, B, and C to better illustrate variation across time points.

- Did any animals exhibit respiratory signs or clinical manifestations related to lung involvement?

- Please specify the viral genome copy number per dose in the inoculum for each RVA strain.

- Consider citing: Hou et al., 2025. "Innate immune sensing of rotavirus by intestinal epithelial cells leads to diarrhea." Cell Host & Microbe 33, 408–419. This study demonstrates that RV-induced diarrhea may occur in the absence of viral replication, suggesting a possible role of innate immune activation by viral particles. It is intriguing whether pathological change in lungs were induced by viral protein or genome lacking infectivity.

- Figure 4: Please add labels showing tissues.

- Related to Figure 4: Please specify which glycan receptors are relevant for each of the RVA strains used in this study.

Reviewer #3: In Fig 1C and S1C, the scale of the y-axis makes it difficult to have a sense for G9 virus titers. Are they five-fold lower than G9? 10-fold? 100-fold? A log2 scale or a linear scale that encompasses a broader range of labeled values (and does not cut off error bars) should be used.

In Fig 2A-B, data are initially presented as shaded grids that depict a range of values. Then, a subset of the results is displayed in bar-graph form. The results chosen for bar graphs might be only the subsets that include data with statistically significant differences. In several cases these subsets include tissues samples for which no viral RNA was detectable. It is also mentioned that lack of statistical significance might derive from high standard deviation, which is reasonable among animals. To increase transparency and share the wealth of information generated by infecting the animals, the authors should display individual data points, overlaid on the bar graphs, for all four time points for G5P[7] and G9P[13]. The same type of data display should be included in the supplement for the other two viruses in the study. Showing each virus on a separate graph might be helpful for appropriate y-axis scaling.

Please add labels for tissue and lectin types to Fig 4A. In Fig 4B, how many independent images were analyzed from each tissue for each lectin type? Did they come from one or multiple animals? Please display the individual data points overlaid on the bar graphs.

In Fig 5, which virus and time point are shown? At minimum, representative images of lungs from animals infected with the viruses most often discussed in the paper (G5 and G9) should be included. Were the features described observed in the lungs of all RVA-infected pigs or just some? If not all, what was the frequency for each virus?

In Fig 6, please indicate the time point and virus shown in each panel. It would be helpful to see representative images from animals infected with G5 and with G9. Higher resolution/zoomed in images are needed to clearly show vacuolar degeneration of enterocytes in the ileum (Fig 6B compared with 6A). What is the rationale for showing H&E staining in the ileum but viral antigen in the jejunum? Were none of the ileum samples virus antigen positive?

How were numbers of animals to be infected for the experiment chosen?

How conserved is the VP6 sequence targeted by ISH probes among the four tested viruses?

Please provide additional details about he CCIF assay used to quantify RVA fecal shedding, including the cell type used for the assay and the antisera used to detect the viruses.

Please describe how working concentrations/dilutions of lectins and antibodies used in the study were selected/optimized

PLOS authors have the option to publish the peer review history of their article (what does this mean? ). If published, this will include your full peer review and any attached files.

**Do you want your identity to be public for this peer review?** For information about this choice, including consent withdrawal, please see our Privacy Policy .

Reviewer #1: **Yes: ** Kyoung-Oh Cho

Reviewer #2: No

Reviewer #3: No

**Figure resubmission:**

**Reproducibility:**



---

## [Decision Letter · Decision Letter 1]

13 Nov 2025

Dear Dr. Vlasova,

We are pleased to inform you that your manuscript 'The efficiency of rotavirus A spread to extraintestinal tissues is not determined by the levels of its replication in the gut' has been provisionally accepted for publication in PLOS Pathogens.

Best regards,

Karl Boehme, PhD

Guest Editor

PLOS Pathogens

Alexander Gorbalenya

Section Editor

PLOS Pathogens

Sumita Bhaduri-McIntosh

Editor-in-Chief

PLOS Pathogens

orcid.org/0000-0003-2946-9497

Michael Malim

Editor-in-Chief

PLOS Pathogens

orcid.org/0000-0002-7699-2064

Reviewer Comments (if any, and for reference):

Reviewer's Responses to Questions

**Part I - Summary**

Reviewer #2: I have carefully reviewed the revised manuscript and appreciate the authors’ thoughtful responses and revisions. The concerns raised in the previous round have been adequately addressed, and the manuscript has been substantially improved in clarity and scientific rigor. I am satisfied with the current version and have no further major comments. I recommend acceptance of the manuscript in its present form.

Reviewer #3: In the current study, Raev et al. orally inoculated germ-free piglets with three porcine (G5P[7], G9P[13], and G4P[6]) and one human (G1P[8]) species A rotavirus (RVA). They hypothesized that RVAs that replicate more efficiently in the gut would spread more efficiently to extraintestinal sites. They characterized viral shedding, the presence of viral RNA, and signs of pathology in the intestine and at sites of dissemination, with a focus on two of the viruses (G5 and G9). Virus titers in intestinal contents and rectal virus shedding were higher for G5 than G9 at some time points (on the order of 4-fold). All viruses were detected in extraintestinal tissues. However, RT-qPCR and RNAish suggested a trend towards more frequent and/or higher levels of extraintestinal dissemination, particularly to the lungs, liver and salivary glands, for G9 than G5. The authors investigated the presence of several glycans known to serve as receptors for some RVAs in various piglet tissues and found the highest levels and diversity of these molecules present in the ileum followed by the lung. Evidence of gross pathology was detected in the lung, histological changes and viral antigen were detected in the lung and gut, and viral antigen was detected in the liver for at least some of the viruses.

Limited numbers of labs possess the capacity to conduct rotavirus infection studies in germ-free piglets. The study found, unexpectedly, that the efficiency of RVA intestinal replication does not necessarily correlate with the efficiency of extraintestinal spread. Although the magnitude of the differences is relatively small, the collective evidence supports a trend toward higher G5 titers in the intestine but higher G9 genome copy numbers at several extraintestinal sites. Thus, these findings suggest that factors other than viral replication influence spread from the primary site of RVA infection. The distribution of glycans in piglet tissues is poorly characterized. Detection of higher levels of several sialylated and non-sialylated glycans that are receptors for some RVAs in piglet ileum and lungs is consistent with the reasonable and testable hypothesis that virus attachment is an important driver of the observed spread and pathology outcomes. Although other factors also may contribute to these differences, this observation opens an avenue for investigation.

**Part II – Major Issues: Key Experiments Required for Acceptance**

Reviewer #2: (No Response)

Reviewer #3: The authors have addressed major concerns sufficiently.

**Part III – Minor Issues: Editorial and Data Presentation Modifications**

Reviewer #2: (No Response)

Reviewer #3: The authors have addressed minor revisions sufficiently.

Line 155-158. Consider whether to describe replication differences as ‘major.’ Many are only several-fold in magnitude.

Fig. 5. By squinting, I realized that there actually are labels on Fig. 5A. I didn’t see them because the text is the same color as the background. Either the background color or text color should be altered for contrast, so that the lectin and tissue labels can be seen.

PLOS authors have the option to publish the peer review history of their article (what does this mean? ). If published, this will include your full peer review and any attached files.

**Do you want your identity to be public for this peer review?** For information about this choice, including consent withdrawal, please see our Privacy Policy .

Reviewer #2: **Yes: ** Yuta Kanai

Reviewer #3: No

---

## [Editor Report · Acceptance letter]

Dear Dr. Vlasova,

We are delighted to inform you that your manuscript, "The efficiency of rotavirus A spread to extraintestinal tissues is not determined by the levels of its replication in the gut," has been formally accepted for publication in PLOS Pathogens.

Best regards,

Sumita Bhaduri-McIntosh

Editor-in-Chief

PLOS Pathogens

orcid.org/0000-0003-2946-9497

Michael Malim

Editor-in-Chief

PLOS Pathogens

orcid.org/0000-0002-7699-2064